# Spontaneous Orofacial Movements at Writhing and Fidgety General Movements Age in Preterm and Full-Term Infants

**DOI:** 10.3390/children9081175

**Published:** 2022-08-05

**Authors:** Regina Donnamaria Morais, Ana Lucia Goulart, Benjamin Israel Kopelman

**Affiliations:** 1Premature Clinic, Escola Paulista de Medicina, Federal University of São Paulo (Unifesp), São Paulo 04023-060, Brazil; 2Neonatal Department, Premature Clinic, Escola Paulista de Medicina, Federal University of São Paulo (Unifesp), São Paulo 04023-060, Brazil; 3Pediatrics, Escola Paulista de Medicina, Federal University of São Paulo (Unifesp), São Paulo 04023-060, Brazil

**Keywords:** child development, general movements, orofacial system, premature infants

## Abstract

Background: As general spontaneous movements at the writhing and fidgety ages have been important for the early identification of neurodevelopmental impairment of both full-term and preterm infants, the knowledge of the spontaneous orofacial movements at these ages also seems to be important for the diagnosis of oral function, particularly in preterm infants. Therefore, we decided to first classify preterm and full-term infants according to general movements ages, and then to record, describe, compare, and discuss their spontaneous orofacial movements. Methods: This cross-sectional study included 51 preterm infants (born between 28 and 36 weeks) and 43 full-term infants who were classified at the writhing and fidgety ages of Prechtl’s method of general movements assessment. Their spontaneous orofacial movements were recorded on video, and The Observer XT software (Noldus) was used to record the quantitative values of the movements. Results: Poor repertoires of writhing movements were more frequent in the preterm infants (90.9%) compared to full-term ones (57.9%). Positive fidgety movements were observed in 100% of both preterm and full-term infants. Oral movements were similar for both preterm and full-term infants, regardless of their movement stage. Conclusion: All spontaneous orofacial movements were present both in preterm and full-term infants, albeit with higher frequency, intensity, and variability at fidgety age.

## 1. Introduction

The physiological skills generally essential for the discharge of preterm infants (PTIs) include sufficient oral feeding to support proper growth, the ability to maintain normal body temperature in a home environment, and sufficient control of a mature respiratory system. Such skills are achieved at different post-menstrual ages [1], depending on birth weight, gestational age at birth, and severity of neonatal diseases. The ability to feed safely involves the functional interaction of the lips, jaw, tongue, palate, pharynx, and esophagus, among other elements of the orofacial system to achieve endurance, performance, strength, efficiency of sucking patterns, and suck–swallow–breathing coordination [2]. These movements have their origin in intrauterine life and are initially quite simple, becoming complex and more coordinated over the course of pregnancy [3,4].

The complexity of oral movements achieved at the end of pregnancy is compromised in PTIs due to underdeveloped oral motor skills and, consequently, uncoordinated suck–swallow–breathing sequencing [5]. This may lead to delays in successful breast- and bottle-feeding, poor weight gain, and dehydration during early postnatal weeks, in addition to changes in the oral experience resulting from necessary interventions during hospitalization [6].

The proper development of intrauterine movements is essential for functional motor readiness after birth, including mobility of the orofacial system to ensure proper feeding function. Birth before full-term age, however, can hinder the evolutionary process of these spontaneous movements (SMs), thus compromising their complexity, particularly for the orofacial system. The diagnosis of oral function is based exclusively on the ability of the orofacial system during sucking, swallowing, and breathing, but there is no record of the spontaneous orofacial system movements in PTIs in the extrauterine environment. Likewise, there are no studies on the relationship between these SMs and the so-called writhing movements (WMs) and fidgety movements (FMs) observed in Prechtl’s general movements assessment (GMA) for the early identification of neurodevelopmental impairment of full-term infants (FTIs) and PTIs [7,8,9].

Considering the lack of literature, based on the need of establishing such a relationship for clinical practice, and in an attempt to encourage further research in this field, the aim of this study was to classify low-risk PTIs and FTIs at the WMs and FMs stages of Prechtl’s GMA, to record the spontaneous orofacial system movements of both PTIs and FTIs, to compare them according to GMA movement state (WMs versus FMs), and to discuss the qualitative observation of such movements.

## 2. Material and Methods

The study included 51 PTIs (born between 28 and 36 weeks of gestational age) and 43 FTIs. According to Prechtl’s GMA, 22 PTIs and 19 FTIs (43.1%) were at the WMs stage and 29 PTIs and 24 FTIs (56.4%) were at the FMs stage. There were no repeated infants in the groups (Figure 1). Throughout a 12-month period in 2017–2018, PTIs and FTIs were recruited, respectively, at the Neonatal Unit and Breastfeeding Clinic of a university hospital. All parents gave their written informed consent for participation.

Exclusion criteria included any condition that could severely compromise the orofacial and the global SM, such as grade III and IV peri-intraventricular hemorrhage, cystic leukomalacia, meningitis, congenital anomalies, genetic syndromes, symptomatic congenital infections at birth, facial paresis, gavage feeding, cleft lip and palate, tongue tie, and ventilatory support or sedative medication at the age of assessment. PTIs who were not discharged before the FMs stage were excluded of the study.

Data regarding maternal schooling and age, type of delivery, and history of twin pregnancy as well as data regarding the infant’s birth were collected in PTIs and FTIs according to WMs and FMs groups.

Spontaneous orofacial system movements were recorded for 3 min, at writhing (37–41 weeks of gestation) and fidgety (9–18 weeks) ages (actual age for FTIs and corrected age for PTIs). FTIs were evaluated after the second day of life, while PTIs were evaluated after discharge.

Spontaneous orofacial system movements and WMs/FMs were recorded separately and simultaneously with two different cameras: a Sony Handheld Camcorder held by the person in charge of the project, in order to follow the movements, even when the child presented head movements and a possible escape from the focus of the image, and an iPhone 6S placed on a tripod at the distance recommended by Prechtl’s method. All infants were placed in supine position, on a contrasting surface, partly dressed, between feedings, and without stimulation, according to Prechtl’s method of GMA.

Video recordings of the spontaneous orofacial system movements were analyzed with The Observer XT software (Noldus) by the first author at the Division of Phoniatric of the Medical University of Graz, Austria. The video files were imported into The Observer XT and quantitative values for the movements were recorded. This software also allowed coding the behavior of each study variable (lip, tongue, commissure) during video playback by the first author, who was duly trained at the mentioned university, and automated calculation of the total number of times, repetitions per minute, and average duration in milliseconds of each movement during the 3 min recording period.

The quality criteria used for the analysis of WMs were those proposed in the classification of Prechtl et al. [10,11]: poor repertoire, cramped synchronized, and chaotic. FMs were classified as normal, abnormal, or absent. The analysis of spontaneous orofacial system movements obeyed the criteria established by our group for this study of the position and movements of the lips (open and closed), tongue (large or tip; symmetrical or asymmetrical) and commissures (symmetrical or asymmetrical) (Figure 2).

## 3. Statistical Analysis

The qualitative variables were described as absolute and relative frequencies. The quantitative variables were described as the mean and standard deviation when normally distributed, or the median (P50) and interquartile range (IQR, P25–P75%) otherwise, as well as minimum and maximum values. To test for the association between qualitative variables, we used the chi-square test or Fisher’s test when expected values were less than 5. To compare groups, we used Student’s *t*-test or the Mann–Whitney test when the data did not show a normal distribution. We used two-way ANOVA to compare the means of quantitative variable (total number of SMs, mean duration, and frequency by minute) changes, according to the levels of two categorical variables: GMs stages (WMs and FMs) and gestational age (PTIs and FTIs). These analyses allowed us to evaluate the interaction of PTIs and FTIs with the Prechtl stage (WMs vs. FMs). The significance level was set at 5%. All statistical analyses were carried out in STATA/SE 15.1 for Windows.

## 4. Results

There were no significant differences between PTIs and FTIs in the WMs group regarding maternal and delivery data. In the FMs group, a higher maternal age and a lower frequency of cesarean delivery were observed in the FTIs group. PTIs had lower weight, length, and head circumference, median Apgar scores, and breastfeeding frequency, as well as a higher frequency of very low and low birth weight (in both WMs and FMs groups). There were no differences regarding sex, skin color, or pacifier use. These data are presented in Table 1.

The GMA revealed poor repertoires in WMs more frequently in PTIs (90.9%) compared to FTIs (57.9%). Positive FMs were observed in 100% of both PTIs and FTIs (Table 2). The records of oral movements were similar when considering all PTIs and FTIs, independently of WMs and FMs ages (Table 3).

On the other hand, there were differences regarding such records when comparing the WMs and FMs groups without considering the gestational age at birth (Table 4). Except for large tongue, the total number and repetitions per minute of all other oral movements were higher, while their mean duration (except for tongue tip) was shorter, in the FMs group compared to the WMs group.

Finally, the comparison of oral movements between PTIs and FTIs when stratified into WMs and FMs groups did not show any differences (Table 5).

Qualitative observation of orofacial system movements at WMs and FMs ages will be presented and commented upon in the Discussion section of this article.

## 5. Discussion

Our findings did not evidence differences between PTIs and FTIs regarding spontaneous orofacial movements. There is no literature to support a typical discussion of such findings, and this is the reason we have used this section to discuss our qualitative observations of spontaneous orofacial movements, so that it can serve as a basis for future linear studies.

Spontaneous orofacial movements and postures of infants at WMs and FMs ages were significantly different. As expected, the differences found are in accordance with Prechtl’s postulation that “the significant transformation from WM to FM age represents a potential biologic function and the ontogenetic adaptation as a calibration of the postnatal proprioceptive system which is necessary for selective, coordinated and intentional movements”. The differences found between WMs and FMs ages are discussed as qualitative observations so that they are better clarified.

The mouth, tongue, and oral commissures play important roles in sensory-motor-oral development, including establishment of the function of the entire digestive system in the infant’s first adaptive responses outside the intrauterine environment [12]. Proper sucking, swallowing, and breathing require mature synchronization of the muscles of suction with the perioral muscles to generate pressure. Mouth opening and closing and tongue mobility are essential for the formation of boluses, and peristalsis for their transport towards the pharynx [13]. Our results showed that low-risk PTIs presented these oral movements just as FTIs do, during both WMs and FMs stages.

The Observer XT software (Noldus) offered not only quantitative data for each orofacial system element, but also an observational record of the most frequent typical global postures and movements (PMs) adopted by infants, at both stages, that influenced orofacial movements.

Regarding WMs, the infants spontaneously adopted the lateral head position, which is typical of newborns’ normal motor development, with upper limbs in external rotation, arms close to the trunk and hands close to the shoulders and head, high chest close to the jaw, and hip, knee, and trunk in physiological flexion. Head movements varied in length, flexion, and midline. We observed kicking movements with lower limbs that facilitated the approach of trunk to chin and hands to face and hand to mouth, by arms flexion. Head rotations are spontaneous movements and are not passively initiated as responses to the asymmetrical cervical tonic reflex (ACTR) [14]. We observed that spontaneous orofacial movements and postures depend on both the quality of typical development global movements and the WMs. 

In the supine position, infants at FMs age presented orofacial movements dissociated from global mobility, and, on the other hand, a wide variety of global PMs, which were guided by the orientation in space and environment through the visual system. It was possible to observe a higher total number of mouth closed even in head extensions or rotations. All oral PMs, however, had short duration due to the alternations of global movements and/or the approach of hands to the mouth resulting from orientation of the limbs in midline. 

### 5.1. Orofacial System Movements at WMs Age

Regarding WMs, we observed less frequency, intensity, and variability of open-and closed-mouth PMs in both PTIs and FTIs compared to FMs age. There was a prevalence of the open-mouth posture due not only to the anatomy of the oral cavity and the shape of the lips [15], but also to the associated movements of the lower lip with the protrusion and retrusion of the tongue. The closed-mouth posture, in the same way, was influenced by the movement of the tongue when in protrusion movement; it favored greater anteriorization of the lower lip and, consequently, contact with the upper lip.

Both the posterior and the anterior excursion of the tongue alternated closed- and open-mouth postures. The mouth opening and closing resulting from the tongue and jaw movement represented an associated movement, in which these structures perform a synchronous, un-dissociated movement in the first months of life [16]. These lip PMs, influenced by the action of the tongue, represented a difference in motor strength between the tongue and lips. This possibly justifies the findings in the function and performance of sucking up to the age of 2–3 months, in which the lips do not exert an effective action in grasping the nipple when sucking. It is known that the lips are positioned slightly supported around the nipple, and the expression of milk occurs through tongue action [6].

Commissure PMs were influenced by typical global movements and reflected in the open-mouth posture. The presence of Moro’s reflex caused an abrupt movement of the entire body and head. In the presence of total extension, a bilateral contraction of the commissures and a unilateral contraction in the return of the head to the midline position were observed. Unilateral contraction was observed when the hand or forearm approached the face, as a response to the tactile stimulus or a search reaction. Almost all commissure PMs were observed in response to a tactile stimulus or as associated reactions.

The observed commissures movement may be related to the lack of balance between flexor and extensor muscles and antigravitational control, as is typical of the neonatal period for an active lateralization movement of the head and weight transfer [17]. Thus, we understand that variation of the commissure posture can be considered a compensatory movement, associated with and facilitated by the change of the head posture through motor actions of the *orbicularis oris* muscle and extrinsic fibers that are inserted in the modiolus, close to the *rima oris* [18].

SMs are a landmark of pre- and postnatal development, playing an adaptive role in development to facilitate the formation of anatomical and sensorimotor systems and the development of motor skills for behavior directed to certain functional objectives [19]. Another explanation would be the dynamic systems theory, which defines the nervous system as goal-oriented, producing emergent and varied motor strategies to meet tasks [20], which, for the orofacial system, are the functions of chewing, social smile, and lip contractions to produce speech sounds.

We observed the presence of a large tongue associated with jaw PMs and varied movement directions in the oral cavity. Despite this variability, it did not seem to be an organized and intentional movement, but a continuity of the movements observed in fetal life, following the three distinct models of tongue movements observed between 15 and 28 weeks of gestation [20]: initially, a protrusion movement towards the lower lip, depression of the tongue in combination with protrusion, and, finally, total protrusion followed by retraction. In the last 10 weeks of gestation, this movement is called the suckling suction movement pattern, with a predominance of tongue movement in the anteroposterior direction and present until the second month of life. We can assume that part of the movements observed herein can be ascribed to the continuum of fetal-life movements. However, the higher position of the tongue can be explained by non-nutritive sucking [21].

Tongue tip was observed during tongue protrusion, accompanied by an extension of the head, trunk, and arms. Some theories can explain these PMs. Regarding the action of tongue protrusion associated with cervical extension, we can assume that it is an adaptive reaction for the maintenance of air space in the oropharynx, since newborns’ oral anatomy shows the tongue anchored in the mandible in a retrognathic position with its base positioned towards the supraglottic passageway. This mandible position and tongue base delimit a relatively small space between the posterior part of the oral cavity and the cervical vertebrae [22].

The movement of symmetry of the tongue and commissure (movements outside the midline of the oral cavity) were observed in the presence of the rooting reflex and head extension with lateral rotation. These movements correspond to those documented by Sheppard and Mysak [23] in their study of oral reflex ontogeny in childhood, in relation to the jaw, lip, and tongue movements in response to external stimuli. After a lateral perioral stimulus, the responses were head rotation, mandibular depression, lower lip deviation, tongue protrusion, and lateralization to the same side of the stimulus. After a stimulus located on the upper right lip, the responses were jaw depression, separation of the lips, tongue protrusion, and lateralization towards the stimulus; a medial lip stimulus resulted in tongue retrusion. These reflex responses around 35 weeks of age were more subtle than at younger ages, involving fewer structures and less movement excursion. This proves the possibility of a variety of oral movements, initially associated with a reflex, which must be modulated in the first months of life, probably through feeding, thus promoting mastery of these motor activities as adaptive responses for the development of the oral-motor-sensory system and enabling functional activity. 

### 5.2. Orofacial System Movements at FMs Age

At the FMs age, we found a higher total number and more repetition of open and closed mouth, tongue tip, and symmetric commissures. However, the average duration of movements was lower for all elements when compared to the same movements at the WMs stage, representing a greater variability of movements, since they were shorter in duration due to a greater alternation between postures.

When the mouth was open, we observed many tongue movements in the superior, anterior, and lateral directions, in a large and tongue tip shape more internalized in the oral cavity and accompanied by commissure movements, with smiles and guttural sounds, demonstrating greater dissociation of oral motor skills.

Other open-mouth PMs were the hand–mouth or hand–feet–mouth movements, exactly as expected for the typical motor development process at 4 months. At this age, there is an increased frequency of the face elements exploration, but with a short duration [17]. Hand–hand exploration was favored by the gain in the midline, that is, bringing the hands together in front of the face, visual contact, and the antigravity positioning of the arms in relation to the trunk [24]. The arms in motion in the antigravity position reach the face more repeatedly, which allows different movements of the mouth, positioning of the commissures, and tongue inside the mouth. These movements and postures of the oral system, however, did not present a fixed pattern.

The closed-mouth posture was maintained even in the sagittal or lateral planes of body movement, and even in the presence of commissure movements. The closed-mouth PMs did not present a contact tonic pressure, which was proved by the tongue action moving anteriorly and the presence of cracks breaking the lip seal [25]. The closing action of the lips assists in the evolutionary process of maturation and physiology of sucking–swallowing, with the movement of raising and lowering the tongue during suction called the “sucking suction” pattern [6,21].

In addition to mouth PMs, we observed SMs with large and tongue tip, positioned centrally or on the side of the oral cavity, and accompanied by symmetric commissures. The large tongue was observed during saliva collection and swallowing, and the tongue tip during exploration of the oral cavity changing the position and commissures movements. The exploration did not represent intentional movement, but a variety of free and independent movements which may represent future articulatory points of speech [26].

When observing commissure symmetry, excluding the tongue action, we observed changes in mouth angle position accompanied by a social smile, both bilateral and unilateral, as reported by Kawakami et al. [27], who registered spontaneous smiles in both PTIs and FTIs at up to 6 days of life and smile and laughter up to 12 weeks, with the duration of bilateral laughter being longer than that of smiles.

Regarding WMs, 90.9% of PTIs and 57.9% of FTIs showed a poor repertoire (that is, a sequence of repetitive movements, large amplitudes, high to moderate speed, and predictive motor behavior, with a loss of variability, frequency, and fluidity of movement). Studies have shown that neonates often present poor repertoires at the WMs stage, which does not necessarily entail abnormal neurodevelopment [28], as it can be transient and change to normal development [29]. In fact, despite these poor repertoires, we observed positive FMs indicative of normal neurological development [10,11,30] in 100% of both PTIs and FTIs.

Our findings regarding spontaneous orofacial system movements at both WMs and FMs ages are in line with the changes in quality from WMs to FMs [31]. This specific age of significant transformation may represent a biological functional potential or an ontogenetic adaptation, such as a postnatal calibration of the proprioceptive system, necessary to achieve adequate coordination of selective and intentional movements [31].

Limitations of this study included the small sample size, so our results should be confirmed in a large group of both PTIs and FTIs. Although the results indicated very clearly that both healthy PTIs and FTIs showed spontaneous orofacial system movements with little variation in number, repetition, and duration, both for the WMs and FMs, it was not a linear and sequenced evaluation. Finally, PTIs included in the study represent a selection of PTIs with the lowest risk for developmental impairment and for this reason cannot represent PTIs in general.

These limitations notwithstanding, the results of this study are of clinical importance, as they indicate that an oral evaluation of healthy PTIs at the corrected age should consider oral responses as linked to the stage of global motor development. Tongue protrusion is a response to maintenance of air passages; great variability of tongue movements represents a continuation of fetal movements; and the frequency of open-mouth movements is a consequence of the conformation of the lips and the size of the oral cavity. At ages between 3 and 5 months, oral responses are dissociated from global movements, with longer lip contact, a more internalized tongue, variation in tongue shape, different points of intraoral contact, and commissure mobility.

## Figures and Tables

**Figure 1 children-09-01175-f001:**
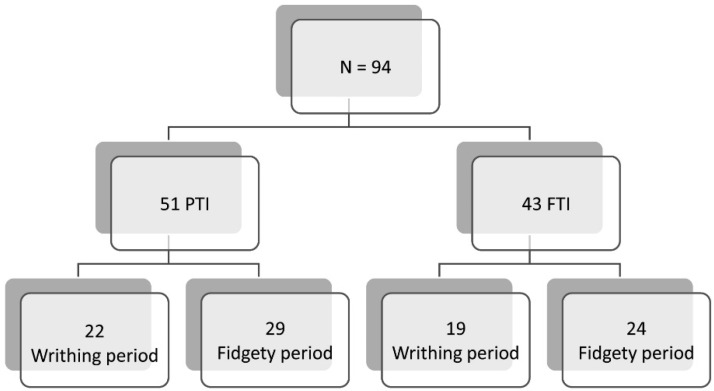
Composition of the four groups of study.

**Figure 2 children-09-01175-f002:**
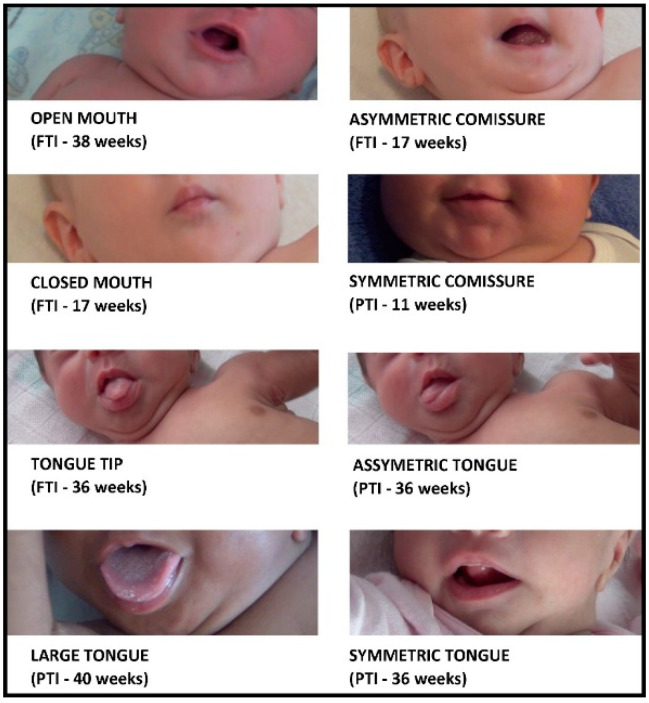
Examples of spontaneous orofacial system movements and postures analyzed in this study.

**Table 1 children-09-01175-t001:** Maternal, birth, and infants’ data according to GMs period and gestational age at birth.

	Writhing			Fidgety		
	Preterm (22)	Full-Term (19)	*p*-Value	Preterm (29)	Full-Term (24)	*p*-Value
Maternal data						
Maternal education (n/%)						
-Incomplete secondary	5/23.8%	5/26.3%	0.782	5/17.9%	1/4.3%	0.202
-Complete secondary	11/52.4%	8/42.1%		16/57.1%	12/52.2%	
-Higher	5/23.8%	6/31.6%		7/25.0%	10/43.5%	
Age (years) *	28.7 ± 8.4%	31.0 ± 7.1	0.350	29.2 ± 8.1	33.3 ± 4.8	0.034
Cesarean delivery (n/%)	19/86.4%	15/83.3%	0.789	27/93.1%	12/50.0%	<0.001
Twin pregnancy (n/%)	6/27.3%	2/10.5%	0.249	9/31.0%	2/8.3%	0.043
Data at birth						
Gestational age (weeks) *	31.8 ± 1.6	38.1 ± 0.9	<0.001	31.5 ± 2.5	39.5 ± 1.0	<0.001
Weight (g) *	1737.5 ± 530.9	2990.3 ± 434.3	<0.001	1591.7 ± 531.1	3372.0 ± 410.2	<0.001
Extreme low weight (n/%)	0 (0.0)	0 (0.0)	-	3 (10.3)	0 (0.0)	0.242
Very low weight (n/%)	7 (31.8)	0 (0.0)	0.010	12 (41.4)	0 (0.0)	<0.001
Low weight (n/%)	13 (59.1)	1 (5.3)	<0.001	10 (34.5)	0 (0.0)	0.001
Length (cm) *	41.0 ± 4.0	47.6 ± 1.9	<0.001	39.5 ± 4.1	48.9 ± 2.8	<0.001
Head circumference (cm) *	29.4 ± 1.8	34.1 ± 1.4	<0.001	28.9 ± 3.2	35.1 ± 2.3	<0.001
Male sex (n/%)	10 (45.4)	9 (47.4)	0.902	18 (62.1)	14 (58.3)	0.782
White (n/%)	10 (45.4)	8 (42.1)	0.829	19 (65.5)	20 (83.3)	0.143
Adequacy for gestational age (GA)						
-Adequate	16 (76.2)	16 (84.2)	>0.999	16 (57.1)	21 (87.5)	0.011
-Small	2 (14.3)	2 (10.5)		10 (35.7)	1 (4.2)	
Apgar (Median/P25–P75%)						
1 min	8/7-8	9/9-9	<0.001	8/6-9	9/8-9	0.032
5 min	9/9-9	10/9-10	0.002	9/9-10	10/9-10	0.016
Feeding						
Breast (n/%)	8 (36.4)	16 (84.2)	0.007	4 (14.3)	17 (70.8)	<0.001
Bottle (n/%)	5 (22.7)	1 (5.3)		18 (64.3)	4 (16.7)	
Others (n/%)	9 (40.9)	2 (10.5)		6 (21.4)	3 (12.5)	
Pacifier users (n/%)	5 (22.7)	3 (15.8)	0.576	13 (44.8)	11 (45.8)	0.942

(*) Mean ± standard deviation.

**Table 2 children-09-01175-t002:** Prechtl’s general movements assessment of all preterm and full-term infants.

Prechtl’s GMA	Preterm Infants (51)		Full-Term Infants (43)		Total		*p*-Value
	n	%	n	%	n	%	
Writhing movements							
-Normal	2	9.1	8	42.1	10	24.4	0.026
-Poor repertoire	20	90.0	11	57.9	31	75.6	
Fidgety movements							
-Normal (F+)	29	100.0	24	100.0	53	100.0	-
-Abnormal (F−)	0	-	0	-	0	-	

**Table 3 children-09-01175-t003:** Records of oral movements (means and standard deviations) for all preterm and all full-term infants.

Oral Movements	Preterm (51)	Full-Term (43)	*p*-Value
Open mouth			
Total number	52.69 ± 30.27	47.46 ± 30.16	0.405
Repetitions per minute	14.62 ± 8.37	13.06 ± 8.87	0.384
Average duration	3.35 ± 2.50	3.43 ± 2.30	0.877
Closed mouth			
Total number	24.42 ± 21.57	26.84 ± 23.42	0.532
Repetitions per minute	6.70 ± 5.75	7.31 ± 6.74	0.570
Average duration	2.63 ± 2.25	3.63 ± 3.20	0.079
Large tongue			
Total number	31.20 ± 18.22	24.77 ± 18.97	0.099
Repetitions per minute	8.60 ± 4.93	6.70 ± 5.17	0.074
Average duration	2.73 ± 2.67	2.58 ± 2.17	0.748
Tongue tip			
Total number	10.06 ± 13.94	8.69 ± 8.73	0.525
Repetitions per minute	2.44 ± 2.82	2.71 ± 3.82	0.644
Average duration	1.22 ± 1.05	1.33 ± 0.78	0.653
Tongue symmetry			
Total number	23.80 ± 20.05	18.36 ± 20.20	0.199
Repetitions per minute	6.53 ± 5.64	5.02 ± 5.57	0.201
Average duration	2.88 ± 3.29	3.48 ± 3.92	0.413
Commissure symmetry			
Total number	29.06 ± 19.30	28.60 ± 19.25	0.936
Repetitions per minute	7.92 ± 4.89	7.88 ± 5.76	0.999
Average duration	2.54 ± 1.48	2.83 ± 1.77	0.364

**Table 4 children-09-01175-t004:** Records of oral movements (means and standard deviations) for infants in the writhing and fidgety movements stages of Prechtl’s general movements assessment.

	Writhing Movements (41)	Fidgety Movements (53)	*p*-Value
Open mouth			
Total number	40.34 ± 25.68	58.00 ± 31.45	0.005
Repetitions per minute	11.06 ± 5.50	16.11 ± 9.87	0.004
Average duration	4.06 ± 2.46	2.85 ± 1.98	0.010
Closed mouth			
Total number	16.62 ± 14.49	32.26 ± 24.88	<0.001
Repetitions per minute	4.58 ± 3.32	8.80 ± 7.20	<0.001
Average duration	3.87 ± 2.85	2.51 ± 2.55	0.017
Large tongue			
Total number	25.10 ± 16.36	30.70 ± 20.21	0.153
Repetitions per minute	6.92 ± 3.92	8.36 ± 5.82	0.175
Average duration	3.12 ± 2.14	2.31 ± 2.61	0.116
Tongue tip			
Total number	6.06 ± 6.30	11.97 ± 13.43	0.019
Repetitions per minute	1.74 ± 1.78	3.24 ± 3.74	0.033
Average duration	1.46 ± 1.05	1.10 ± 0.83	0.102
Tongue symmetry			
Total number	16.72 ± 15.11	24.96 ± 22.86	0.052
Repetitions per minute	4.59 ± 3.83	6.84 ± 6.57	0.057
Average duration	4.06 ± 4.33	2.45 ± 2.72	0.032
Commissure symmetry			
Total number	20.39 ± 14.48	35.40 ± 19.90	<0.001
Repetitions per minute	5.54 ± 2.78	9.73 ± 6.00	<0.001
Average duration	3.49 ± 1.87	2.05 ± 1.03	<0.001

**Table 5 children-09-01175-t005:** Means and standard deviations of the total number, repetitions per minute, and average duration (milliseconds) of oral movements according to GMs stage and gestational age at birth.

	Writhing			Fidgety		
	Preterm (22)	Full-Term (19)	*p*-Value	Preterm (29)	Full-Term (24)	*p*-Value
Open mouth						
Total number	46.18 ± 28.75	33.58 ± 20.27	0.169	57.62 ± 30.95	58.46 ± 32.48	0.917
Repetitions per minute	12.49 ± 5.42	9.40 ± 5.23	0.237	16.22 ± 9.85	15.97 ± 10.12	0.909
Average duration	4.04 ± 2.31	4.09 ± 2.69	0.934	2.81 ± 2.15	2.90 ± 1.82	0.895
Closed mouth						
Total number	17.05 ± 17.31	16.16 ± 11.02	0.895	29.76 ± 23.01	35.29 ± 27.16	0.347
Repetitions per minute	4.60 ± 3.58	4.56 ± 3.13	0.980	8.23 ± 6.55	9.49 ± 8.00	0.441
Average duration	3.19 ± 1.97	4.67 ± 3.51	0.086	2.23 ± 2.38	2.84 ± 2.76	0.402
Large tongue						
Total number	29.86 ± 17.67	19.58 ± 13.07	0.079	32.21 ± 18.88	28.88 ± 21.99	0.515
Repetitions per minute	8.18 ± 3.92	5.45 ± 3.46	0.086	8.91 ± 5.62	7.70 ± 6.10	0.385
Average duration	3.05 ± 1.79	3.19 ± 2.55	0.854	2.49 ± 3.19	2.10 ± 1.73	0.557
Tongue tip						
Total number	6.95 ± 7.10	4.86 ± 5.03	0.572	10.08 ± 9.76	14.94 ± 17.69	0.162
Repetitions per minute	2.00 ± 1.96	1.39 ± 1.49	0.550	2.79 ± 2.82	3.94 ± 4.87	0.234
Average duration	1.37 ± 1.18	1.60 ± 0.86	0.491	1.10 ± 0.94	1.09 ± 0.65	0.979
Tongue symmetry						
Total number	20.82 ± 16.27	11.72 ± 12.20	0.153	26.07 ± 22.51	23.56 ± 23.72	0.652
Repetitions per minute	5.63 ± 3.93	3.31 ± 3.37	0.190	7.22 ± 6.64	6.37 ± 6.58	0.584
Average duration	3.56 ± 4.31	4.72 ± 4.41	0.309	2.36 ± 2.17	2.56 ± 3.33	0.844
Commissure symmetry						
Total number	22.77 ± 18.25	17.63 ± 7.89	0.359	33.83 ± 18.99	37.29 ± 21.21	0.483
Repetitions per minute	6.01 ± 3.05	5.01 ± 2.40	0.515	9.38 ± 5.53	10.16 ± 6.62	0.565
Average duration	3.40 ± 1.67	3.60 ± 2.12	0.663	1.90 ± 0.91	2.23 ± 1.15	0.412

## Data Availability

Data of this study are available upon request to the authors due to privacity maintenance.

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
