# Peer review of "Spontaneous Orofacial Movements at Writhing and Fidgety General Movements Age in Preterm and Full-Term Infants"

_children, 2022, doi:10.3390/children9081175_

Round 1
Reviewer 1 Report
Reviewer Comments to Editor
This is an interesting topic and the authors have attempted to characterize a little-known aspect of spontaneous movements in preterm and full-term infants. It also includes a very detailed account of orofacial movements and their possible functional purpose.
Reviewer Comments to Author
General comments:
While the topic was interesting, it was difficult to understand the technology that underpins the quantitative characterization of the spontaneous orofacial movements.
Specific comments:
Abstract:
“As general spontaneous movements at the writhing and fidgety ages have been important for early identification of risks for the neurodevelopment” – I would change to “As general spontaneous movements at the writhing and fidgety ages have been important for early identification of neurodevelopmental impairment”
Introduction:
“Prechtl’s General Movements Assessment (GMA) for the early identification of risks for the neurodevelopment of full-term infants” – same comment as above
It might be helpful to insert a hypothesis based on the existing literature as this seems not to be studied systematically as yet.
Methods:
“The study included 51 PTI (born between 28 and 36 weeks gestational age) and 43 FTI. According to Prechtl’s GMA, 41 infants (43.1%) were at the WM stage and 53 (56.4%) at the FM stage.” – Were some infants included twice in the study? (i.e. at the WM and FM stages?)
“while PTI were evaluated after discharge” – does this mean that some PTI were excluded if they were not discharged before the FM stage?
“a Sony Handheld Camcorder and an iPhone 6S” – could you specify if these cameras were in fact “hand-held” or if they were placed at the recommended 1 m distance at 45 degrees as per the Precthl method?
“Observer XT-Noldus software at the Research Unit of Interdisciplinary Developmental Neuroscience of a foreign medical university” – it would be good to know about blinding (i.e. were those analyzing these movements blinded to the infants’ course etc.). Also, which “foreign medical university”?
“This software also allowed coding the behavior of each study variable (lip, tongue, commissure) during video playback and calculation of the total number of times, repetitions per minute, and average duration in milliseconds of each movement during the 3-minute recording period.” – is this an automated technology? Or does it require an individual to go through frames and record movements?
“The analysis of SOFSM obeyed the criteria established by this study of position and movements of the lips (open and closed), tongue (large or tip; symmetrical or asymmetrical) and commissures (symmetrical or asymmetrical)” – do you have a reference for these criteria or were they developed by your group?
Results:
A flow chart may be helpful to understand how patients were ultimately recruited (e.g. if there were repeated patients).
Discussion:
Overall, this section seems to need a bit more grounding in existing literature. There were several areas where references would have been helpful. I have detailed some below.
“It seemed to us very relevant to observe SOFSM in both WM and FM.” – while I agree, I wonder if this statement should be based on literature to suggest that oral movements are somehow connected to feeding, which I am not sure that there is literature to suggest this.
I wonder if some diagrams may help us understand the parts of the mouth/face that were observed and some of the qualitative and quantitative variables collected.
How does the presence of tongue tie affect some of these orofacial movements? Was data collected on tongue tie/tongue tie releases in this population or were these children excluded?
Tables and Figures:
Table 2 – there was quite a high rate of poor repertoire in the full-term group (57.9%). Is there a possible reason for this? Were these infants with significant breastfeeding difficulty as well?
Table 5 – why are those numbers bolded if there were no significant differences (p > 0.05 in both cases)?
Author Response
Response to Reviewer 1 Comments
Point 1: “As general spontaneous movements at the writhing and fidgety ages have been important for early identification of risks for the neurodevelopment” – I would change to “As general spontaneous movements at the writhing and fidgety ages have been important for early identification of neurodevelopmental impairment”
Response: Thank you for the recomendation. It was included in the article (abstract).
Point 2: Prechtl’s General Movements Assessment (GMA) for the early identification of risks for the neurodevelopment of full-term infants” – same comment as above.
Response: Also changed as recommended in Introduction (at the end of the 4th paragraph.
It might be helpful to insert a hypothesis based on the existing literature as this seems not to be studied systematically as yet.
Response: In fact, there is no literature on orofacial movements and postures in the Prechtl’s periods of general movements to support a hypothesis for the study. Our objective with this research was exactly to identify orofacial movements in both Prechtl’s periods in preterm and full-term infants. We changed the last paragraph of the Introduction including not a hypothesis itself, but the reasons that led us to develop the study.
Point 3: Were some infants included twice in the study? (i.e. at the WM and FM stages?)
Response: No one was included twice in the study. We clarified the information in the first paragraph of Material and Methods.
Point 4: “while PTI were evaluated after discharge” – does this mean that some PTI were excluded if they were not discharged before the FM stage?
Response: Yes. We include this information as an exclusion criterion in the second paragraph of Material and Methods.
Point 5: “a Sony Handheld Camcorder and an iPhone 6S” – could you specify if these cameras were in fact “hand-held” or if they were placed at the recommended 1 m distance at 45 degrees as per the Precthl method?
Response: We added the information in the 5th paragraph of Material and Methods. The Sony Handheld Camcorder was not placed as recommended in Prechtl’s method because of the head movements of infants leading to possible escapes from the focus on mouth movements and loss of records.
Point 6 : “Observer XT-Noldus software at the Research Unit of Interdisciplinary Developmental Neuroscience of a foreign medical university” – it would be good to know about blinding (i.e. were those analyzing these movements blinded to the infants’ course etc.). Also, which “foreign medical university”?
Response: We clarified this point and included the name of the University etc. in the 6th paragraph of Material and Methods. In fact, the first author conducted all the observations, records, and analysis, therefore not blinded to the infants’ course. Should it be considered a limitation for the study.
Point 7: “This software also allowed coding the behavior of each study variable (lip, tongue, commissure) during video playback and calculation of the total number of times, repetitions per minute, and average duration in milliseconds of each movement during the 3-minute recording period.” – is this an automated technology? Or does it require an individual to go through frames and record movements?
Response: In fact, it is a hybrid procedure. We clarified it in the same 6th paragraph of Material and Methods. Coding the behavior during video playback was carried out by the first author after long training. Calculations were an automated procedure.
Point: “The analysis of SOFSM obeyed the criteria established by this study of position and movements of the lips (open and closed), tongue (large or tip; symmetrical or asymmetrical) and commissures (symmetrical or asymmetrical)” – do you have a reference for these criteria or were they developed by your group?
Response 8: These criteria were developed by our group considering the well-established knowledge that these movements planes of the tongue, commissures and lips are important to future function to eat and speech.
Point 9: “It seemed to us very relevant to observe SOFSM in both WM and FM.” – while I agree, I wonder if this statement should be based on literature to suggest that oral movements are somehow connected to feeding, which I am not sure that there is literature to suggest this.
Response: We removed this paragraph from discussion in attendance of the comments of the second reviewer. Anyway, references 12 and, specially, 13 support the connection between oral movements and feeding.
Point 10: How does the presence of tongue tie affect some of these orofacial movements? Was data collected on tongue tie/tongue tie releases in this population or were these children excluded?
Response: Sorry. Frenulum of the tongue was an exclusion criterion. We added this lost information in Materials and Methods.
Point: Table 2 – there was quite a high rate of poor repertoire in the full-term group (57.9%). Is there a possible reason for this? Were these infants with significant breastfeeding difficulty as well?
Response: We believe that the reason of this high rate of poor repertoires in the full-term infants is clear in Discussion and is supported by references 27 and 28. There was no breastfeeding difficulties in our full-term infants.
Table 5 – why are those numbers bolded if there were no significant differences (p > 0.05 in both cases)?
Response. It was an error of typing. It was corrected in the Table.

Reviewer 2 Report
Major Concerns:
1. The discussion spends most the time discussing the qualitative observation and not much on the difference between groups. Maybe consider changing the title/aim or rewrite most of the discussion to reflect the aims.
2. Too many abbreviations! I highly recommend less than 5 abbreviations in a paper. Otherwise, it’ll make your paper really difficult to follow. You do not need abbreviations for spontaneous movements or SOFSM…etc
3. Statistics: You can use a two-way ANOVA when you have data on a quantitative dependent variable at multiple levels of two categorical independent variables, are you considering duration as categorial data?Please clarify.
4. Can you explain the algorithm for the Observer XT-Noldus software analysis? Was this automated? Or required manually coding? If it’s manual, did you have personas reliability trained and what’s the agreement score?
5. Maybe add some figures/photos/videos to show the behaviors you described. It’s hard to picture them in mind.
6. Discussion: First paragraph of the discussion normally should be a summarization of the results. The current first paragraph is more like an intro for me.
7. 4th paragraph of discussion, (We believe that the behaviors observed can be justified by spontaneous head rotation), WM are justified by head rotation? Please clarify what behavior?
8. 5th paragraph of discussion, how is alert status and visual interaction with the environment a global posture and movements? And how is this paragraph help explain your results? what's its correlation with OFS?
9. (OFS movements at WM ag ). I did not see this outcome mentioned in the results section. If this is important for the discussion, maybe mention them in the result section as qualitative results.
1. (OFS movements at FM age), 1st paragraph, change in frequency is not necessary change in variability, please clarify
1. OFS movements at FM age), 3rd paragraph: Not clear. Why arms further from the trunk and visual exploration of the hands cause increasing of frequency and decreasing of duration? I don't understand the connection. Please clarify.
1. Please focus more on the difference between group or why there is no difference and what’s previous studies indication for your outcomes?
Minor revisions:
1. 4th paragraph of discussion, (kicks movements) should be kicking movements
2. 4th paragraph of discussion, (The presence of this response allowed us to observe) What response?
3. (OFS movements at WM ag), 1st paragraph, (Regarding WM, we observed less frequency, intensity) , “less” comparing to what?
Author Response
Response to Reviewer 2 Comments
Point 1: The discussion spends most the time discussing the qualitative observation and not much on the difference between groups. Maybe consider changing the title/aim or rewrite most of the discussion to reflect the aims.
Response: We choose to include the discussion of the qualitative observations in the aim of the study (Abstract ans the end of Introduction). In fact, this article is a extract of the Doctorat Thesis of the first author and the original research aimed at a deep discussion of the qualitative observations.
Point 2: Too many abbreviations! I highly recommend less than 5 abbreviations in a paper. Otherwise, it’ll make your paper really difficult to follow. You do not need abbreviations for spontaneous movements or SOFSM…etc
Response: We thank you for the recommendation. Some abbreviations were excluded.
Point 3: Statistics: You can use a two-way ANOVA when you have data on a quantitative dependent variable at multiple levels of two categorical independent variables, are you considering duration as categorial data?Please clarify.
Response: We do not know if we have understood your point. We tried to clarify changing the information: “We used two-way ANOVA to compare means of quantitative variables (total number of SM, mean duration, and frequency by minute) changes, according to the levels of two categorical variables: GM stages (WM and FM) and gestational age (PTI and FTI).”
Point 4: Can you explain the algorithm for the Observer XT-Noldus software analysis? Was this automated? Or required manually coding? If it’s manual, did you have personas reliability trained and what’s the agreement score?
Response: We clarified this information this way: “Observer XT and quantitative values for the movements were recorded. This software also allowed coding the behavior of each study variable (lip, tongue, commissure) during video playback by the first author, who was duely trained at the mentioned university, and automated calculation of the total number of times, repetitions per minute, and average duration in milliseconds of each movement during the 3-minute recording period”.
Point 5: Maybe add some figures/photos/videos to show the behaviors you described. It’s hard to picture them in mind.
Response: A figure was included in the article, as follows:
Point 6: Discussion: First paragraph of the discussion normally should be a summarization of the results. The current first paragraph is more like an intro for me.
Response: We changed the first paragraph of Discussion: “Our findings regarding spontaneous orofacial movements in WM and FM ages did not evidence differences between PTI and FTI. There is no literature to support a typical discussion of such findings, and this is the reason we have used this section to discuss our qualitative observations of spontaneous orofacial movements, so that it can serve as a basis for future studies. Significant differences found out between WM and FM ages are discussed as qualitative observations are presented so that they are better clarified”.
Point 7: 4th paragraph of discussion, (We believe that the behaviors observed can be justified by spontaneous head rotation), WM are justified by head rotation? Please clarify what behavior?
Response: We excluded this information.
- 5thparagraph of discussion, how is alert status and visual interaction with the environment a global posture and movements? And how is this paragraph help explain your results? what's its correlation with OFS?
Response: The information was completed for better understanding: “In the supine position, infants at FM age presented orofacial movements dissociated from global mobility, and, on the other hand, a wide variety of global PMs which was guided by the orientation in space and environment through the visual system. It was possible to observe higher total number of mouth closed even in head extensions or rotations. All oral PMs, however, had short duration due to the alternations of global movements and/or the approach of hands to the mouth resulting from orientation of the limbs in midline”.
- (OFS movements at WM ag). I did not see this outcome mentioned in the results section. If this is important for the discussion, maybe mention them in the result section as qualitative results.
Response: It is now explained at the end of Results: Qualitative observation of orofacial system movements at WM and FM ages will be presented and commented in the Discussion of this article.
- (OFS movements at FM age), 1stparagraph, change in frequency is not necessary change in variability, please clarify
Response: We are not sure to have understood your point. We excluded the word “frequency” and replaced it for “higher total number and more repetition”. In fact, the greater the total number and the repetition, the greater the variability of the movements, and vice versa. Did we clarify your point?
- OFS movements at FM age), 3rdparagraph: Not clear. Why arms further from the trunk and visual exploration of the hands cause increasing of frequency and decreasing of duration? I don't understand the connection. Please clarify.
Response: We added the following information: “The antigravity position of the arms in motion reached the face more times, changing faster both the position of commissures or tongue and the closed and open mouth postures. Eye-head contact in midline changing head posture and lip position was also more frequent and faster. These movements and postures of the oral system, however, did not present a fixed pattern.
- Please focus more on the difference between group or why there is no difference and what’s previous studies indication for your outcomes?
Response: In fact, we judge relevant describe and discuss qualitative observations to better understand the differences or no differences. Furthermore, there is no previous studies on orofacial system movements and postures in WG and FM for indicating our outcomes.
Minor revisions:
- 4thparagraph of discussion, (kicks movements) should be kicking movements
Response: It was corrected. Thank you.
- 4thparagraph of discussion, (The presence of this response allowed us to observe) What response?
Response: We excluded the paragraph. It was really not clear.
- (OFS movements at WM ag), 1stparagraph, (Regarding WM, we observed less frequency, intensity) , “less” comparing to what?
Response: To FM. We added the information.

Round 2
Reviewer 1 Report
Thank you for all your edits, I think the paper reads much better and addresses the concerns from the previous review. I appreciate the figure as well explaining some of the orofacial movements that are being analyzed. I do, however, note that there was one comment that was not addressed:
Results:
A flow chart may be helpful to understand how patients were ultimately recruited (e.g. if there were repeated patients).
As well, I think it needs one more read through and editing for language errors.
Author Response
Thank you for all your edits, I think the paper reads much better and addresses the concerns from the previous review. I appreciate the figure as well explaining some of the orofacial movements that are being analyzed. I do, however, note that there was one comment that was not addressed:
Results:
A flow chart may be helpful to understand how patients were ultimately recruited (e.g. if there were repeated patients).
RESPONSE: There were no repeated patient, as we stated in the last response. Anyway, we added a flow chart in Methods.
As well, I think it needs one more read through and editing for language errors.
RESPONSE: We checked and made some corrections. The original manuscript was revised by a native English speaker. If the case, he could revise it again considering the adjustments we have done, but it would take more than the five days we have had to ressubimit it.
Reviewer 2 Report
1. I was hoping to see more discussion on the difference between group or why there is no difference. I understand there is not much literature to support your points but it would be interesting to state your hypothesis and to stimulate the conversation for future studies.
2.Discussion:1st paragraph: 'to discussing'-'to discuss'
3.Next sentence, remove 'found out'
4.Discussiong:4th paragraph: This paragraph basically explained WM which is defined in GMA. I think the point is what does this study add to what we already know? And how does it relate to OFS? Originally you were saying WM allows you to observe more behaviors, but how does WM affect OFS?
5.Orofacial system movements at FM age: 3rd paragraph: 'both the position of commissures or tongue and the closed and open mouth postures changed faster.'
6.Next sentence: eye-hand contact? Also confusing sentence structure and grammar mistakes.
7.The figures are really helpful.
Author Response
- I expected to see more discussion about the difference between the groups or why there is no difference. I understand that there is not much literature to support your points, but it would be interesting to expose your hypothesis and stimulate conversation for future studies.
RESPONSE: We changed the two first paragraphs of Discussion in an attempt to clarify your comments:
Our findings did not evidence differences between PTI and FTI regarding spontaneous orofacial movements. There is no literature to support a typical discussion of such findings, and this is the reason we have used this section to discuss our qualitative observations of spontaneous orofacial movements, so that it can serve as a basis for future linear studies.
Spontaneous orofacial movements and postures of infants at WM and FM ages were significantly different. As expected, the differences found are in accordance with Prechtl’s postulates that “the significant transformation from WM to FM age represents a potential biologic function and the ontogenetic adaptation as a calibration of the postnatal proprioceptive system which is necessary for selective, coordinated and intentional movements”.
- Discussion:1st paragraph: 'to discussing'-'to discuss'
RESPONSE: OK
3.Next sentence, remove 'found out'
RESPONSE: OK
- Discussion: 4th paragraph: This paragraph basically explained WM which is defined in GMA. I think the point is what does this study add to what we already know? And how does it relate to OFS? Originally you were saying WM allows you to observe more behaviors, but how does WM affect OFS?
RESPONSE:
This paragraph describes the classification of spontaneous movements and postures at WM period that are typical of normal motor development. It is important to better understand our observations regarding OFS movements. It is not about the WM itself, but about the typical movements of normal development at WM age. In other words, we could observe the quality of the normal development global movements beyond the little amplitude and velocity described at WM age, and the dependence of orofacial movements on such global movements of normal development at WM age (which is different of what occurs at FM when movements are dissociated).
We completed the paragraph adding:
“We observed that spontaneous orofacial movements and postures depend on both the quality of normal development global movements and the WM.”
5.Orofacial system movements at FM age: 3rd paragraph: 'both the position of commissures or tongue and the closed and open mouth postures changed faster.'
RESPONSE: OK
6.Next sentence: eye-hand contact? Also confusing sentence structure and grammar mistakes.
RESPONSE:
We changed the sentence:
There was an increase in the frequency and velocity of eye-hand exploration in midline, resulting in increased frequency of different movements of mouth, position of commissures and the tongue inside the mouth.
7.The figures are really helpful.